# Effects of continuous positive airway pressure on neurocognitive architecture and function in patients with obstructive sleep apnoea: study protocol for a multicentre randomised controlled trial

Huajun Xu,[1,2,3] Hui Wang,[1,2,3] Jian Guan,[1,2,3] Hongliang Yi,[1,2,3] Yingjun Qian,[1,2,3] Jianyin Zou,[1,2,3] Yunyan Xia,[1,2,3] Yiqun Fu,[1,2,3] Xinyi Li,[1,2,3] Xiao Jiao,[1,2,3] Hengye Huang,[4] Pin Dong,[5] Ziwei Yu,[5] Jun Yang,[6] Mingliang Xiang,[6] Jiping Li,[7] Yanqing Chen,[7] Peihua Wang,[8] Yizhou Sun,[8] Yuehua Li,[9] Xiaojian Zheng,[10] Wei Jia,[10,11] Shankai Yin[1,2,3]

► Prepublication history and additional material is available. To view please visit the journal (http://dx.doi.org/10.1136/bmjopen-2016-014932).

HX and HW contributed equally.

For numbered affiliations see end of article.

**Correspondence to**
Jian Guan;
guanjian0606@sina.com and
Hongliang Yi;
yihongl@126.com

## ABSTRACT

**Objectives** Many clinical studies have indicated that obstructive sleep apnoea (OSA), the most common chronic sleep disorder, may affect neurocognitive function, and that treatment for continuous positive airway pressure (CPAP) has some neurocognitive protective effects against the adverse effects of OSA. However, the effects of CPAP treatment on neurocognitive architecture and function remain unclear. Therefore, this multicentre trial was designed to investigate whether and when neurocognitive architecture and function in patients with OSA can be improved by CPAP treatment and to explore the role of gut microbiota in improving neurocognitive function during treatment.

**Methods/design** This study will be a multicentre, randomised, controlled trial with allocation concealment and assessor blinding. A total of 148 eligible patients with moderate to severe OSA will be enrolled from five sleep centres and randomised to receive CPAP with best supportive care (BSC) intervention or BSC intervention alone. Cognitive function, structure and function of brain regions, gut microbiota, metabolites, biochemical variables, electrocardiography, echocardiography, pulmonary function and arterial stiffness will be assessed at baseline before randomisation and at 3, 6 and 12 months.

**Ethics and dissemination** This study has been approved by the Medical Ethics Committee of Shanghai Jiao Tong University Affiliated Sixth People's Hospital (approval number 2015-79). The results from this study will be published in peer-reviewed journals and at relevant conferences.

**Trial registration number** NCT02886156; pre-results.

## Strengths and limitations of this study

► Multicentre, randomised, prospective study.
► Relatively large number of participants (n=148).
► Strictly standardised study population.
► The multicentre pragmatic design strengthens future results and generalisability.
► Possible selection bias may exist, as not all possible patients may enter in the study.

obstruction during sleep.[1] There is a high prevalence (2%–4%) of this condition in adults aged 30–60 years, which is increasing with the growing epidemic of obesity.[2–4] OSA is well recognised as a risk factor for neurocognitive, cardiovascular and metabolic disorders.[5–7] It is worth noting that neurocognitive dysfunction is associated with a series of sequelae, such as social and economic burdens.[8–10] Continuous positive airway pressure (CPAP) is typically recommended as a first-line treatment for OSA, as it can partly or completely reverse OSA-related complications, including cognitive impairments.[11 12] In addition, CPAP may be effective for reversing structural and functional brain changes.[13 14]

To date, only two randomised controlled trials (RCTs) have simultaneously evaluated the effects of CPAP on cognition and neuroimaging; both studies had relatively small sample sizes and short intervention periods.[15 16] However, the response to CPAP treatment in the trials was variable. Thus, the relationships among OSA, brain and

## BACKGROUND

Obstructive sleep apnoea (OSA) is a common chronic sleep disorder characterised by recurrent partial or complete pharyngeal

neurobehavioural function are complex and must be comprehensively evaluated, including how, when and how long CPAP treatment should be applied for effective therapy.

The gut microbiota is an important factor that affects neurological functions[17] and may be involved in the modulation of central nervous function (eg, the hypothalamus–pituitary–adrenal axis) and associated with neuropsychiatric conditions.[17 18] The composition of the gut microbiota is altered through intermittent hypoxia or sleep fragmentation in animal models.[19 20] However, there have been no studies to determine whether changes in human intestinal microbiota are associated with OSA and whether or how the microbiota can be modulated by CPAP. This study will clarify the complex relationships among OSA, gut microbiota and neurological functions.

## METHODS/DESIGN

### Study aims and hypotheses
The primary aim of this study is to assess the dynamic (3, 6 and 12 months) effects of CPAP on cognitive function, neurocognitive architecture and function. The primary hypothesis of this study is that, compared with the non-CPAP treatment group, patients with OSA who receive CPAP intervention will show improved cognitive function through structural and functional brain modulation immediately after 3 months of CPAP treatment, and that these benefits will persist at the 6-month and 12-month follow-ups. The secondary aims of this study are to investigate whether and which sleep parameters are correlated with improvements of cognitive function, neurocognitive architecture and function; to explore whether the metabolic status assessed by metabolomics can be reversed by CPAP and correlated with improvement of neurocognitive impairment; and to investigate whether CPAP modulates the gut microbiota and contributes to alleviation of neurocognitive impairment. We hypothesise that patients with OSA who undergo CPAP treatment will have metabolic improvement and modulation of gut microbiota, and that these changes will have beneficial effects on neurocognitive impairment.

### Study design
The Shanghai Multicenter Obstructive Sleep Apnea Therapy trial is a multicentre, evaluator-blinded, randomised, parallel-controlled trial performed in five sleep centres (Department of Otolaryngology and Head and Neck Surgery, Shanghai Jiao Tong University Affiliated Sixth People's Hospital; Department of Otolaryngology-Head and Neck Surgery, Shanghai First People's Hospital, Shanghai Jiao Tong University; Department of Otolaryngology, Renji Hospital, School of Medicine, Shanghai Jiao Tong University; Department of Otolaryngology, Head and Neck Surgery, Xinhua Hospital, Shanghai Jiao Tong University School of Medicine; Department of Otolaryngology, The Ninth People's Hospital Affiliated to Shanghai Jiao-Tong University

School of Medicine) in the Shanghai district. The design of this protocol strictly followed the Standard Protocol Items: Recommendations for Interventional Trials[21] and the Consolidated Standards of Reporting Trials statements.[22] The study will enrol 148 eligible participants with moderate to severe OSA from the five above-mentioned sleep centres and randomise them to receive CPAP with best supportive care (BSC) intervention or BSC intervention alone. Cognitive function, structure and function of brain regions, metabolites, gut microbiota, biochemical variables, electrocardiography, echocardiography, pulmonary function and arterial stiffness will be assessed at baseline before randomisation and at 3, 6 and 12 months. The flow diagram of this RCT is presented in figure 1.

### Study population
The study population (30–65 years old) will be enrolled from Shanghai and the surrounding area, and all of the participants will be from the Han population. The diagnostic, inclusion and exclusion criteria for this trial will be as described below.

### Diagnostic criteria
According to the American Academic Sleep Medicine (AASM) 2007 criteria,[23] moderate and severe OSA are defined as an apnoea–hypopnoea index (AHI) ≥15 and 30 events per hour, respectively, and the normal state is defined as AHI <5 events per hour.

### Eligible participants
#### Inclusion criteria
1. Informed consent
2. Age 30–65 years
3. Newly diagnosed OSA (full-night in-laboratory polysomnography (PSG) with an AHI ≥15 events per hour)
4. Ability to tolerate CPAP treatment based on 1 night of use (must use it for at least 4 hours)
5. No participation in any other clinical trial in the past 3 months
6. Able to accomplish relevant tests and follow-up

#### Exclusion criteria
1. Severe systemic diseases (eg, cardiac, hepatic and renal failure)
2. Psychiatric conditions (eg, depression, mania, schizophrenia)
3. Neurological diseases (eg, head trauma, brain tumour, epilepsy, stroke, transient ischaemic attack, coma)
4. Sleep disorders other than OSA (narcolepsy, insomnia, chronic sleep deprivation, rapid eye movement (REM) behaviour disorder and restless legs syndrome, central sleep apnoea (≥50% of AHI composed of central apnoeas) or obesity hypoventilation syndrome)
5. Alcoholism (according to the National Institute on Alcohol Abuse and Alcoholism, alcoholism is defined as alcohol consumption exceeding 14

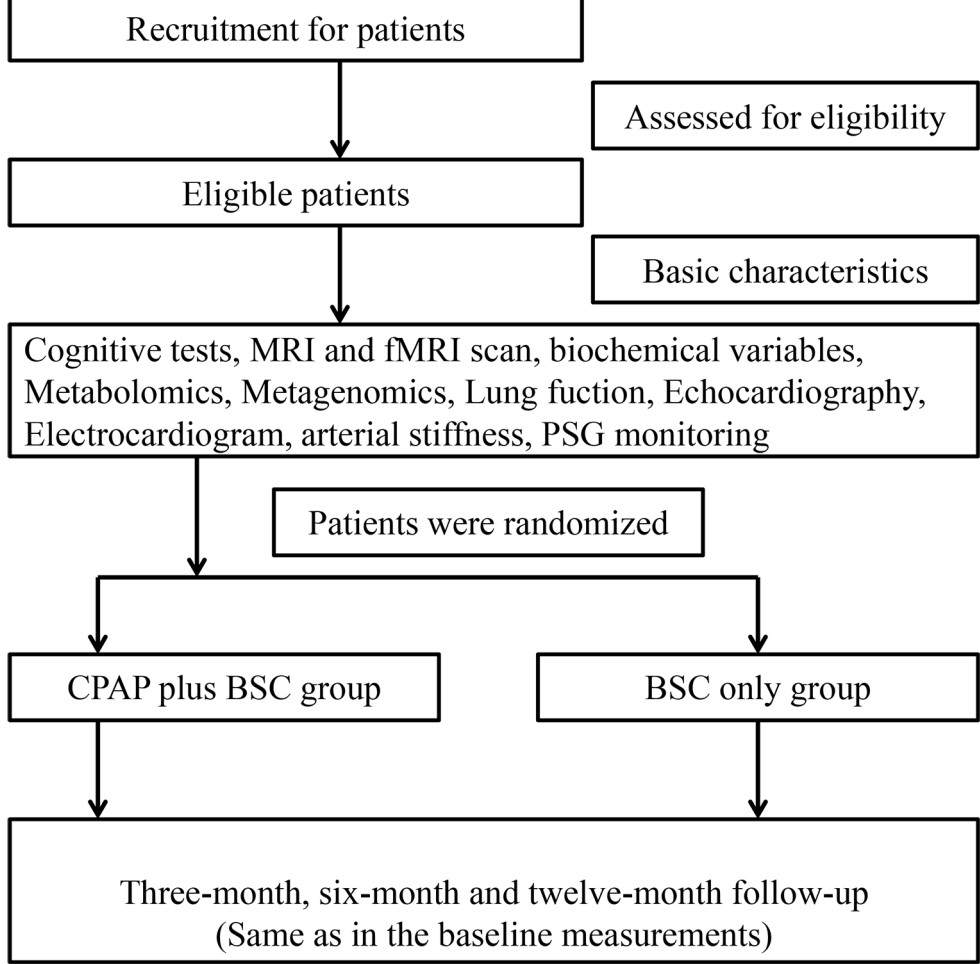

**Figure 1** The flow diagram of this randomised controlled trial. BSC, best supportive care; CPAP, continuous positive airway pressure; fMRI, functional MRI; PSG, polysomnography.

standard drinks per week or four drinks per day in men or exceeding seven standard drinks per week or three drinks per day in women. A standard drink was defined as one 12-ounce bottle of beer, one 5-ounce glass of wine or 1.5 ounces of distilled spirits),[24] drug addiction, use of psychotropic drugs, current sedative use or narcotics

6. Prior therapy for OSA (ie, CPAP, upper airway surgery, oral appliance)
7. Mini Mental State Examination (MMSE) score <24
8. Left handed
9. MRI contraindications (eg, claustrophobic or metal implantation)
10. Gastrointestinal surgery during the last year, except for appendicitis and hernia surgery
11. Pregnancy
12. Use of intestinal flora regulator (eg, antibiotics or probiotics) in the previous 8 weeks
13. Medical treatment for cholecystitis, gallstones, gastrointestinal ulcers, urinary tract infection, acute pyelonephritis or cystitis in the past 3 months
14. Infectious diseases, such as tuberculosis, AIDS

15. Commercial drivers or people deemed to be at risk of driving-related accidents
16. Deemed by the researchers to be unsuitable for this trial

## Withdrawal criteria
Subjects will be permitted or required to withdraw from this study if they have insurmountable claustrophobia or other unbearable conditions associated with CPAP use, major protocol violation or the development of serious systemic diseases that prevents continuation in this study.

## Recruitment and screening
Participants will be recruited from Shanghai and its vicinity through newspapers and instant messaging software (WeChat). Potential participants with self-reported severe snoring and with an interest in participating will be screened by a standard PSG and examined for eligibility (AHI ≥15 times per hour) for multicentre RCT. Eligible participants will be given a detailed information about this trial.

## Randomisation, allocation concealment and blinding

All eligible patients with moderate to severe OSA will be randomly and equally allocated (randomisation ratio 1:1) to the CPAP plus BSC group or BSC only group. The central random allocation sequence will be produced using the statistical software SAS V.9.4 and managed by an independent statistician (from the Department of Epidemiology, School of Public Health, Shanghai Jiao Tong University) who will be blinded to the participants' recruitment, evaluation and intervention. The research assistant from each sleep centre will request (via email) a participant's allocation from the statistician. The group allocation will not be blinded by participants or intervention supervisors in this trial; however, the patients' group assignments can be blinded to result assessors and statisticians (ie, which patients are allocated to the CPAP treatment versus BSC groups will not be disclosed).

## Intervention

### BSC only group

Participants in the BSC only group will receive advice regarding lifestyle modification, sleep hygiene, naps, exercise, caffeine and diet and avoiding alcohol consumption, but no specific weight loss programme, diet or salt restriction will be suggested.

### CPAP plus BSC group

Participants in the CPAP plus BSC group will receive CPAP treatment plus the aforementioned BSC intervention. CPAP treatment (LOTUS AUTO; Curative Medical Technology, Beijing, China) will be initiated using standard clinical practice (before CPAP treatment, the standard clinical practice is for the doctors to educate patients on the use of CPAP and provide positive encouragement) at each centre. Online monitoring of CPAP compliance can also be used to improve compliance. Participants in both groups will be asked to continue their usual medical care during the trial.

## Follow-up

At 3, 6 and 12 months during the intervention period, all of the participants will undergo the same examinations as at baseline. During the follow-up period, CPAP usage information will be automatically downloaded daily from auto CPAP via the Internet.

## Outcome assessment

The variables in this trial consist of basic characteristics, primary outcomes and secondary outcomes. The basic characteristics will be measured at baseline, and primary and secondary outcomes will be measured at baseline and at the end of each intervention period (3, 6 and 12 months after randomisation). All of the primary and secondary outcomes will be assessed by experienced staff at each sleep centre who will be blinded to treatment allocation.

## Basic characteristics

Participants' basic characteristics (eg, age, sex, years of education, medical history, medication use and occupation) will be gathered by recruiters using a questionnaire. The two standardised neurobehavioural scales, such as Montreal Cognitive Assessment (MoCA) and MMSE scores will be assessed. The assessors in each sleep centre will be trained by a neurologist. Before randomisation, all baseline assessment will be finished.

## Primary and secondary outcomes

### Primary endpoints

MoCA.

### Secondary endpoints

MMSE.

Default mode network and cortical thickness are measured by functional and structural MRI.

Cerebral metabolites, such as N-acetylaspartate (NAA), choline (Cho), creatine (Cr), glutamate and glutamine and myo-inositol are measured by magnetic resonance spectroscopy (MRS).

Neuropsychological tests will also be performed, including mental arithmetic, memory scanning, movement perception, switching attention, space location memory span and attention allocation choice reaction time curve fit.

Sleep variables include AHI, oxygen desaturation index, lowest oxygen saturation, mean oxygen saturation, micro-arousal index, total sleep time, total REM sleep time, total slow wave sleep time, etc.

### Auditory testing

Central auditory processing tests performed include speech-in-noise test, competing sentences test, gap detection and dichotic listening.

Data were recorded using 256-channel high-density electroencephalograph (EEG) recordings.

The gut microbial community will be quantified using metagenomics analysis.[25] Low-molecular-weight metabolites (≤1000 kDa) will be quantified by metabolomics analysis.[26]

Amyloid-β protein 42, presenilin 1, amyloid-β protein 40, total GSK-3β and serine-9 phosphorylated GSK-3β

### Other outcomes

#### Biochemical variables

Biochemical variables include lipid profile (total cholesterol, triglycerides, high-density lipoprotein cholesterol, low-density lipoprotein cholesterol, apoA-I, apoB, apoE and lipoprotein(a)), glucose, insulin, HbA1c, free fatty acids, cholinesterase, lipopolysaccharide-binding protein, toll-like receptor-4, TLR-2, hepatic function tests (alanine aminotransferase, aspartate aminotransferase, γ-glutamyl endopeptidase, alkaline phosphatase, total protein, albumin, total bilirubin, direct bilirubin, bile acid, prealbumin), renal function tests (creatinine, urea nitrogen, uric acid), microalbuminuria and apolipoprotein E genotyping.

### Inflammatory biomarkers

Inflammatory biomarkers include leptin, regulated on activation, normal T cell expressed and secreted, monocyte chemotactic factor 1, interleukin (IL)-6, IL-8, tumour necrosis factor-α, hypersensitive C-reactive protein, resistin, serum amyloid A protein and α-acid glycoprotein.

### Anti-inflammatory biomarkers

Anti-inflammatory biomarkers include adiponectin and IL-1Ra.

### Oxidative stress index

Oxidative stress index may include: 8-isoprostane, blood pressure (systolic blood pressure and diastolic blood pressure), anthropometric variables (weight, height, waist, hip, chest and neck circumferences, visceral fat area and subcutaneous fat area), echocardiography, arterial stiffness (pulse wave velocity, ankle brachial index and toe-brachial index), lung function (eg, forced expiratory volume in 1 s and forced vital capacity) and ECG.

## Study examination procedures

All of the study measurements will be performed before and after 3, 6 and 12 months of CPAP intervention at each centre, except for lung function and ECG.

### Anthropometric measurements

Height and weight will be measured using standard methods. BMI (weight (kg)/height$^2$ (m$^2$)) will be calculated from height and weight. Waist, hip and neck circumferences will be measured using a measuring tape in the conventional manner.

### Urine samples

First midstream urine samples will be collected in the morning with the collected urine samples stored immediately at −80°C for further analyses.

### Blood samples

Fasting blood samples will be collected in the morning, and the samples will be immediately sent for further analyses such as biochemical analysis.

### Faecal samples

Faecal samples will be collected and stored at −80°C until use in subsequent assays. The composition of microbiota will be measured using 16S rRNA-hybridisation and DNA staining.

### Sleep respiratory variables

Laboratory-based PSG (Alice 4 or 5, Respironics; Pittsburgh, PA, USA) will be used to record sleep parameters. EEG, electrooculography, ECG, electromyography, nasal/oral airflow, thoracic/abdominal movement, pulse oximetry, body posture and snoring will be recorded. The definition of OSA is established according to the AASM 2007 criteria.

### Neuropsychological tests

Two standardised questionnaires (MoCA and MMSE) will be administered. Neuropsychological tests will also be performed, including mental arithmetic, memory scanning, movement perception, switching attention, space location memory span and attention allocation choice reaction time curve fit.

### Auditory testing

Audiograms will be measured in one-octave steps at frequencies ranging from 125 Hz to 8 kHz and then at 10, 12.5, 16 and 20 kHz. DPOAE values will be recorded in both groups using a Madsen Capella cochlear emissions analyser in a soundproof room. Only patients with normal audiometry (threshold below 25 dB HL at all frequencies from 0.125 to 20 kHz), normal DPOAE levels, normal tympanometric curve of type A and presence of ipsilateral and contralateral stapedial reflexes will be included.

### Central auditory processing testing

The following central auditory processing tests will be performed in a soundproof room. Before the formal test, participants will be allowed to practice as many times as required and provide feedback to become familiar with the stimuli. (1) Speech-in-noise test: all of the signal processing will be performed with MATLAB software. The speech of 20 ten-word sentences will be presented under both quiet and noisy conditions with speech-shaped noise and will be delivered bilaterally through Sennheiser HD205 headphones at a signal:noise ratio of −5 dB. (2) Competing sentences test: babble noise and the target sentence will be presented bilaterally at an intensity level of 65 dB SL (above speech reception threshold) at a signal:noise ratio of −3 dB. (3) Gap detection: the purpose of this evaluation is to determine the smallest interval that the listener can detect. Two stimuli separated in time by a 'gap' will be presented, and the listener will be asked to detect the gap, essentially discriminating sound from silence. Stimuli as white noise will be bilaterally presented as background noise at intensities 40 dB higher than the hearing threshold. (4) Dichotic listening: two different numbers will be simultaneously presented to both ears with the subjects instructed to repeat both numbers in any order. The test includes 20 number pairs presented dichotically. The number pairs will be aligned for onset and offset to eliminate any cue for the first number heard. Dichotic listening performance will be scored as the percentage of correctly reported different numbers from the right and left ears in the noise-exposed and control groups.

### 256-Channel high-density EEG recording

EEG data will be recorded with 256 channels on an EGI HydroCel Geodesic Sensor Net, with Cz used as the reference channel. All of the electrode impedances will be detected and maintained at less than 50 kΩ. The oddball stimuli pattern with speech sound is used in this study (via Eprime V.2.0; Psychology Software Tools). The

auditory oddball task includes presentation sound of /Ba/, whereas /Da/ is presented as the 'oddball' target stimulus. The stimulus /Ba/ is presented in 85% of the trials together with the stimulus /Da/ in 15% of the trials. The whole task consists of a total of 1000 auditory stimuli with random interstimulus intervals ranging from 850 ms to 1450 ms. The sound stimuli with an intensity of 75 dB delivered through two loudspeakers at a distance of 100 cm from the subjects. During the P3 cortical waves test, the subjects will be asked to press a switch held in the right hand each time they detect a target tone with the eyes closed. In addition, they will be asked not to respond to any auditory stimulus while watching silent films during the MMN test. Source analysis of event-related potentials as MMN, describing neural sources of measured scalp potentials, will be estimated with GeoSource V.2.0.

### Brain imaging

MRI will be performed using the Siemens 3T Verio scanner (MAGNETOM, Siemens) with a 12-channel coil using the same parameters. High-resolution T1-3D-weighted and T2-3D-weighted MRI sequences will be conducted for all subjects. *T1-3D axial images*: field of view (FOV), 230 mm; repetition time (TR)/time to echo (TE), 1500/2.96 ms; flip angle, 9°; voxel size, $0.9 \times 0.9 \times 1$ mm$^3$; slice thickness, 1 mm; and distance factor, 50%. *T2-3D axial images*: FOV, 230 mm; TR/TE, 1500/2.96 ms; voxel size, $0.9 \times 0.9 \times 1$ mm$^3$; slice thickness, 1 mm; and distance factor, 50%. In the same position, a s-fMRI BOLD sequence will be acquired in the rest condition, and 240 functional images will be obtained (repetition time 2000 ms, echo time 30 ms, thickness 4.0 mm, gap 1.2 mm, acquisition matrix 64×64, flip angle 90°, field of view $230 \times 230$ mm$^2$, 30 axial slices with gradient-recalled echo-planar imaging pulse sequence covering the whole brain). Patients will be asked to keep their eyes closed and not to focus their thoughts on any particular topic or to fall asleep. Diffusion kurtosis imaging will be performed in all subjects using 30 gradient directions, six b values and an echo-planar diffusion-weighted imaging sequence. FOV, 300 mm; TR/TE, 1100/109 ms; matrix size 128×128; and slice thickness 4 mm.

### Cerebral metabolites

MRS will be performed using the Siemens 3T Verio scanner with a 12-channel coil. A $2.0 \times 2.0 \times 2.0$ cm$^3$ voxel of interest will be placed at the frontal lobe. NAA, Cho and Cr will be acquired by a MEGA-PRESS sequence using the following parameters: TE/TR=135/5000 ms; average number 80 (scan time 8 min).

### Abdominal and liver fat measurements

The hepatic, subcutaneous and visceral fat will be measured by axial scanning of the upper abdomen (from the diaphragm to the lower edge of the liver). The participants will be instructed to hold their breath to reduce the artefacts induced by respiratory movement. Anatomical T1-weighted spin-echo MRI will be obtained using the

following parameters: slice thickness 5 mm; slice spacing 2.5 mm.

### Safety measurements

Any unexpected adverse events associated with CPAP intervention during OSA-treatment period will be monitored and then reported. The severity and causality of adverse events should also be analysed. The data we submit will be monitored by the Ethics Committee of Shanghai Jiao Tong University Affiliated Sixth People's Hospital. If serious adverse events occurred, they should be immediately reported to the aforementioned ethics committee.

### Sample size calculation

A previous study indicated that the mean and SD of MoCA score are 26.1 (2.0) and 27.4 (2.3) before and after CPAP treatment, respectively.[27] With a significance level of 0.05 and statistical power of 0.9, the calculated sample size is 59 for each group. Considering a 20% loss to follow-up, a total of 148 participants will be required for randomisation.

### Statistical methods

The statistical analysis in this trial will be performed using intention-to-treat (ITT). For missing data, a multiple imputation method will be performed. Normality of the data distribution will be examined by the Kolmogorov-Smirnov test. Continuous variables will be expressed as mean±SD or median (first to third quartile) for normal distribution data or non-normal distribution data, respectively. Categorical variables will be expressed as percentages. To assess primary and secondary endpoints in this trial, a mixed-effects linear model under restricted maximum likelihood and a logistic regression model will be used to analyse the effect of CPAP treatment on the continuous or categorical variables, respectively. In each model, the independent variables should include group (CPAP plus BSC group and BSC group), confounding factors (eg, age, BMI and waist:hip ratio, etc), measurement time (eg, baseline and 3-month, 6-month and 12-month follow-ups) or any other inequality factors between groups at baseline assessment. When a significant main effect was observed, a *post hoc* comparison for different time points (eg, baseline and 3-month, 6-month and 12-month follow-ups) or between groups (CPAP plus BSC group and BSC group) will be performed using multiple comparisons. All of the statistical analyses will be performed using SAS V.9.4, with a two-sided p value<0.05 indicating a statistical significance.

### Data collection and management

All of the data collectors and evaluators will be blinded to the group assignments. The data in this trial will be collected using traditional print-based case report forms, which will then be transcribed by a research assistant into web-based case report forms (w-CRFs) with an oracle clinical (OC)/remote data capture (RDC) system (https://clinicaldata.shsmu.edu.cn/). The OC/RDC system will

be provided by the School of Public Health of Shanghai Jiao Tong University (Shanghai, China) and will meet the standards of security. Data in w-CRFs will be stored in the OC/RDC system at each of the five research centres separately with password protection. The database incorporates automatic data checking (ie, logic/inaccurate ranges and maintains any changes to the data).

### Ethical issues and dissemination

This protocol will be performed according to the Declaration of Helsinki. Ethics approval (Ethics number: 2015-79) has been obtained from Shanghai Jiao Tong University Affiliated Sixth People's Hospital. Before enrolment, until a full discussion about risks and potential benefits has been conducted with each participant, written informed consent will be obtained. This study will be published in peer-reviewed journals and presented as oral presentation or poster at conferences in the field of neurology and sleep medicine. Study data will be made available to researchers, healthcare providers and professionals who are on request with the appropriate human research ethics committees' approvals.

## DISCUSSION

Increasing numbers of clinical studies have shown that CPAP is an effective therapeutic method for neurocognitive dysfunction in OSA. However, the effects and duration of CPAP treatment on neurocognitive function have not been clearly established. This study focuses on the effects of CPAP on middle-aged patients with moderate to severe OSA. Through a 3-month, 6-month and 12-month intervention with CPAP treatment, the results from a range of clinical and biochemical assessments will provide a clearer information on the effects of treatment in OSA.

Measurements of neurocognitive function before and after intervention in this trial may strengthen evidence for links between OSA and neurocognitive dysfunction or reveal new associations. Through CPAP intervention, it will be possible to explore the mechanism of action of CPAP on neurocognitive function (including structural and functional effects), allowing us to investigate the causal relationships between OSA and neurocognitive dysfunction.

The microbial community structure in rodents was altered when the rodents were subjected to intermittent hypoxia or sleep fragmentation.[19 20 28] In addition, sleep fragmentation-induced metabolic alterations may be mediated by concurrent changes in the gut microbiota.[28] Consequently, gut microbiome-targeted therapeutics may help reduce the major end-organ morbidities of OSA. Through metagenomics and metabolomics approaches, we will determine whether gut microbiota play a pivotal role in OSA-related complications, especially in alleviating neurocognitive dysfunction.

To our knowledge, the proposed study will be the first to investigate the effects of CPAP on neurocognitive dysfunction in OSA and to explore the underlying mechanisms using combined '-omics' approaches. The results of this multicentre trial may be helpful for improving the neurocognitive function of patients with OSA.

This protocol describes a multicentre RCT to assess the effects of CPAP treatment on neurocognitive function in patients with moderate to severe OSA. To reduce potential bias, multiple rigorous methods, such as randomisation, blinding of raters and statistical analysers and ITT principles, will be implemented. Potential limitations of this protocol should also be addressed. First, sham CPAP is not used as a placebo; therefore, neither participants nor trial researchers will be blinded to the protocol. Thus, performance bias may be inevitable. Second, although all of the raters have been trained by the same neurologist, there may be heterogeneity in neuropsychological tests in each centre.

In summary, this will be the first multicentre RCT to dynamically observe the effects of CPAP on neurocognitive function and to explore the role of the gut microbiota in neurocognitive dysfunction in OSA. The results of this study will help to clarify potential mechanisms for alleviation of OSA and improvement of neurocognitive dysfunction.

### Trial status

At the time of manuscript submission, the enrolment of volunteers is ongoing.

**Author affiliations**
[1]Department of Otolaryngology Head and Neck Surgery and Center of Sleep Medicine, Shanghai Jiao Tong University Affiliated Sixth People's Hospital, Shanghai, China
[2]Otolaryngological Institute of Shanghai Jiao Tong University, Shanghai, China
[3]Clinical Research Center, Shanghai Jiao Tong University School of Medicine, Shanghai, China
[4]Department of Epidemiology, School of Public Health, Shanghai Jiao Tong University, Shanghai, China
[5]Department of Otorhinolaryngology, Shanghai First People's Hospital, Shanghai Jiao Tong University, Shanghai, China
[6]Department of Otolaryngology Head and Neck Surgery, Xinhua Hospital, Affiliated to Shanghai Jiaotong University School of Medicine, Shanghai, China
[7]Department of Otorhinolaryngology Head and Neck Surgery, Ren Ji Hospital, School of Medicine, Shanghai Jiao Tong University, Shanghai, China
[8]Department of Otolaryngology Head and Neck Surgery, Shanghai Ninth People's Hospital, Affiliated to Shanghai Jiaotong University School of Medicine, Shanghai, China
[9]Department of Radiology, Shanghai Jiao Tong University Affiliated Sixth People's Hospital, Shanghai, China
[10]Shanghai Key Laboratory of Diabetes Mellitus and Center for Translational Medicine, Shanghai Jiao Tong University Affiliated Sixth People's Hospital, Shanghai, China
[11]Cancer Epidemiology Program, University of Hawaii Cancer Center, Honolulu, Hawaii, USA

**Contributors** XH, WH and YS conceived the study, designed the study protocol and drafted the manuscript. XH, WH, GJ, YH, QY, ZJ, XY, FY, LX, JX, DP, YZ, YJ, XM, LJ, CY, WP, SY, LY, JW and YS participated in the coordination and implementation of the study. XH, WH and YS revised and finalised the study protocol. YS is in charge of coordination and direct implementation. HH helped to develop the study measures and analyses. All authors contributed to drafting the manuscript and have read and approved the final manuscript. The corresponding authors have full access to all the data in the study and have final responsibility for the decision to submit for publication.

**Funding** This study was supported by grants-in-aid from multicentre clinical research project from the School of Medicine, Shanghai Jiao Tong University (DLY201502).

**Disclaimer** The study sponsor played no role in the design, methods, data management or statistical analysis or in the decision to publish.

**Competing interests** None declared.

**Provenance and peer review** Not commissioned; externally peer reviewed.

**Data sharing statement** No additional unpublished data are available.

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
