## [Reviewer comments · BMJ Open]

ARTICLE DETAILS

TITLE (PROVISIONAL)	Effects of Continuous Positive Airway Pressure on Neurocognitive Architecture and Function in Patients with Obstructive Sleep Apnea: Study Protocol for a Multicenter Randomized Controlled Trial
AUTHORS	Xu, Huajun; Wang, Hui; Guan, Jian; Yi, Hongliang; Qian, Yingjun; Zou, Jianyin; Xia, Yunyan; Fu, Yiqun; Li, Xinyi; Jiao, Xiao; Huang, Hengye; Dong, Pin; YU, Ziwei; Yang, Jun; Xiang, Mingliang; Li, Jiping; Chen, Yanqing; Wang, Peihua; Sun, Yizhou; Li, Yuehua; Zheng, Xiaojian; Jia, Wei; Yin, Shankai

VERSION 1 - REVIEW

REVIEWER	Lian Leng Low Singapore General Hospital, Singapore Health Services, Singapore
REVIEW RETURNED	12-Nov-2016

GENERAL COMMENTS	This is a well written study protocol deserving of consideration for publication. However, the following points should be addressed. Therefore, my decision would be major revision. 1. The study title should include the study design.2. Page 7 lines 16 - 21: The comparison group should be more clearly articulated in the primary hypothesis.3. Page 9 line 52: I believe it should be "mini mental state examination" instead of "minimum mental state examination".4. Page 9 line 55: The exclusion of left handed patients should be elaborated on.5. page 10 line 17-18: Typo. Should be "unsuitable" instead of "suitable".6. Page 11 lines 1 - 24: The randomization process requires clarifications. Is it 1:1 ratio? Any blocks? Stratification by centre? What do the authors mean by equally allocated? How do the study investigators ensure the outcome assessors are blinded?7. Page 11 line 46: "standard clinical practice" needs to be defined. What is standard practice? Will the study investigators be checking the fidelity and compliance to this standard practice?8. Page 12 line 5: What is auto CPAP?9. Page 12 line 51: additional ")" after brain regions
--

	10. Page 13 line 27: typo for "approch" 11. Page 18 safety measurements: Is there a data monitoring committee for this trial? 12. Page 18 sample size calculation: How was the sample size of 74 for the health controls derived? 13. Page 21: Could the authors discuss the pros and cons of not using sham CPAP in this study? 14. The SPIRIT checklist needs to be attached.
--	--

REVIEWER	Craig L Phillips University of Sydney, Australia
REVIEW RETURNED	18-Nov-2016

GENERAL COMMENTS	This proposed multi-center RCT aims to determine whether and when neurocognitive architecture and function in OSA patients can be improved by CPAP treatment. The authors hypothesise that 3 months of CPAP treatment of OSA will result in improved cognitive function through "structural and functional brain modulation" and that this improvement will continue at 6 and 12 month follow-ups. Hence the primary aim is to determine whether CPAP treatment improves neurocognitive function and "neurocognitive architecture". A number of secondary aims include investigating whether changes in sleep parameters are associated with cognitive function, neurocognitive architecture and function and whether metabolomics and gut microbiota changes have beneficial effects on neurocognitive impairment. The 5 center study plans to adopt a prospective randomized open label blinded endpoint (PROBE) design and intends to follow both SPIRIT and CONSORT statements. The interventions are CPAP + Best Supportive Care (BSC) or BSC alone. All outcomes will be assessed at Baseline, 3, 6 and 12 months. A control group who are presumably OSA free will also be enrolled. Questions and comments for the investigators:  1. Study design: What is the rationale for embedding a case-control study in this RCT? 2. Study design: Only patients who are CPAP compliant (inclusion criteria) will be included. It was unclear when compliance was to be determined – before or after randomization? If determined before, then will patients be washed out prior to randomizing? If after, then this will introduce bias into the study since CPAP compliance may also be a surrogate for compliance with other medications and lifestyle practices. 3. Diagnostic criteria: AASM defined AHI is mentioned – please reference which AASM criteria will be used? 4. Inclusion criteria: Age 30-65 – please explain why adults younger than 30 and older than 65 are excluded? 5. Inclusion criteria: Newly diagnosed OSA – do you mean newly diagnosed severe OSA? By excluding moderate OSA the findings will be less generalizable. 6. Inclusion criteria: Adherence to CPAP treatment – what is
---

considered adherent?

7. Exclusion criteria: Sleep disorders - Central sleep apnea – please define what proportion of the AHI needs to be central before patients are excluded?

8. Exclusion criteria: Alcoholism – please define?

9. Exclusion criteria: Sedatives – is this present use or any (past or present) use?

10. Exclusion criteria: Deemed by the researchers to be “suitable” for this trial – do you mean unsuitable? Will you include commercial drivers or people at risk of driving related accidents?

11. Withdrawal criteria: Adverse events related to CPAP – please define?

12. Intervention: CPAP treatment will be initiated using “standard clinical practice”. Please clarify what this involves?

13. Intervention: How will the effectiveness of CPAP treatment be determined?

14. Sleep variables: Why are there no sleep parameters being measured – eg total sleep time, total REM sleep time and total slow wave sleep time.

15. Outcome Assessment: (page 12, line 21) – the staff will be “blinded to the allocation results of participants”. Do you mean staff will be blinded to participant treatment allocation?

16. Primary outcome: Please clarify what the primary outcome is? According to the manuscript, several are listed (MoCA and MMSE as well as “structure” and “function” of related brain regions using fMRI - where neither structure nor function is clearly defined).

17. Primary outcome: It is also unclear whether this list of outcomes is a composite outcome or not. If it is a composite outcome then please define how it will be determined?

18. Secondary outcome: Please describe with references the Metagenomics and Metabolomics analysis and what will be quantified?

19. Brain imaging: The past tense is used as if this imaging has already taken place? For example: “MRI scan was conducted”, “24 functional images were obtained”. Please clarify?

20. Cerebral metabolites: The same past tense “scans were performed”.

21. Abdominal and liver fat: “were measured”.

22. Abdominal fat: There is no mention of where the images will be scanned in relation to bony landmarks – for example abdominal fat might be quantified from contiguous slices starting at the L4/5 intervertebral space. Please clarify?

23. Statistics: It is unclear what statistical analysis will be performed to assess primary and secondary outcomes.

24. Discussion: It is mentioned that the investigators will determine whether gut microbiota “play a pivotal role in OSA”. This implies that alterations in microbiota somehow cause OSA? Please explain.

25. Discussion: “The results of this study will help to clarify potential mechanisms for alleviation of OSA” – please explain how this might occur?

Summary of main points:
Overall this is an interesting study but relatively ambitious in regards to the large number of investigations and outcomes that are to be collected at multiple time-points. There will likely be a large burden on patients and one wonders whether the projected 20% loss to follow-up is realistic.

It is unclear why the investigators have opted to embed a case-control study into this randomized controlled trial.

	It is of concern that only patients who are compliant with CPAP will be eligible as this will introduce bias if compliance is assessed post randomization. It is unclear how the primary outcome with multiple variables is to be assessed. The age range and OSA severity of patients will limit generalizability of findings.
--	--

REVIEWER	Caroline Mitchell Academic Unit of Primary Medical Care, Faculty of Medicine, Dentistry and Health, University of Sheffield, United Kingdom
REVIEW RETURNED	19-Nov-2016

GENERAL COMMENTS	This protocol paper fulfils the criteria for reporting randomised controlled trials: a CONSORT flow chart will be generated. Analysis must state 'intention to treat analysis', which it does not currently. I would comment that this study is collecting an extraordinary range of secondary outcome data and I would be concerned about participant outcome measurement burden. In order to address this, it would be helpful to know which of the outcome measures are actually part of routine care in this healthcare context and which are being performed as additional research measures. For example, I am concerned about the additional 'study within a study': the exploration of whether 'CPAP modulates the gut microbiota and contributes to alleviation of neurocognitive impairment'. We hypothesize that patients with OSA who undergo CPAP treatment will have metabolic improvement and modulation of gut microbiota, and that these changes will have beneficial effects on neurocognitive impairment'. As a secondary outcome arm of the proposed main study, this would be theory generating and the study is not powered to assess any such associations. I would therefore consider this part of the protocol to be over and above the outcome measures needed to answer the main research question. Whilst the study protocol authors may have funding in place for this additional work, I do not consider that this arm of the study should necessarily be published as integrated within the main study protocol.
--

VERSION 1 – AUTHOR RESPONSE

Reviewer: 1

Reviewer Name: Lian Leng Low

Institution and Country: Singapore General Hospital, Singapore Health Services, Singapore

Please state any competing interests: None declared.

Please leave your comments for the authors below

This is a well written study protocol deserving of consideration for publication. However, the following points should be addressed. Therefore, my decision would be major revision.

Comment 1. The study title should include the study design.

Response 1: Thank you very much for your comment. We added the study design into the title

according to your suggestion.

Original:

Effects of Continuous Positive Airway Pressure on Cognitive Function, Neurocognitive Architecture and Function in Patients with Obstructive Sleep Apnea: The Shanghai Multicenter Obstructive Sleep Apnea Therapy Trial

Revised:

Effects of Continuous Positive Airway Pressure on Cognitive Function, Neurocognitive Architecture and Function in Patients with Obstructive Sleep Apnea: Study Protocol for A Multicentre, Randomised, Controlled Trial

Comment 2. Page 7 lines 16 - 21: The comparison group should be more clearly articulated in the primary hypothesis.

Response 2: Thank you very much for your comment. We stated the comparison group in the primary hypothesis as you suggested.

Original:

The primary hypothesis of this study is that patients with OSA will show improved cognitive function through structural and functional brain modulation immediately after 3 months of CPAP treatment, and that these benefits will continue at the 6-month and 12-month follow-ups

Revised:

The primary hypothesis of this study is that, compared with the non-CPAP treatment group, patients with OSA who receive CPAP intervention will show improved cognitive function through structural and functional brain modulation immediately after 3 months of CPAP treatment and that these benefits will continue at the 6-month and 12-month follow-ups.

Comment 3. Page 9 line 52: I believe it should be "mini mental state examination" instead of "minimum mental state examination".

Response 3: Thank you very much for your comment. We revised "mini mental state examination" instead of "minimum mental state examination" according to your suggestion.

Original:

Minimum Mental State Examination (MMSE)< 24

Revised:

Mini Mental State Examination (MMSE)< 24

Comment 4. Page 9 line 55: The exclusion of left handed patients should be elaborated on.

Response 4: Thank you very much for your comment. The dominant brain hemisphere differs between left-handers and right-handers. In our study, participants will receive fMRI and MRS at some specific position of brain. the results of neurocognitive architecture and function might be affected if we do not exclude the left-handed patients. As most of people are right-handed in China, thus we choose right-handed ones..

Comment 5. page 10 line 17-18: Typo. Should be "unsuitable" instead of "suitable".

Response 5: Thank you very much for your comment. We revised "unsuitable" instead of "suitable" according to your suggestion.

Original:

15) Deemed by the researchers to be suitable for this trial

Revised:

15) Deemed by the researchers to be unsuitable for this trial

Comment 6. Page 11 lines 1 - 24: The randomization process requires clarifications. Is it 1:1 ratio? Any blocks? Stratification by centre? What do the authors mean by equally allocated? How do the study investigators ensure the outcome assessors are blinded?

Response 6: Thank you very much for your comment. 1) the randomization is 1:1 ratio. 2) We have

no blocks for we have relatively large number of participants in this study. 3) We will use central randomization. The random number will be sent by Email from a statistician from Department of Epidemiology, School of Public Health, Shanghai Jiao Tong University. Participants who participant in this study at each of five sites will be randomised into the trial without any blocks. 4) Equally allocated means that the randomization is 1:1 ratio. 5) The outcome assessors will be blinded to the patients' group assignments (i.e., nobody will tell them which patient is allocated to CPAP treatment group or BSC group).

Original:

After baseline assessment, eligible patients with severe OSA will be randomly and equally allocated to the CPAP plus BSC group or BSC only group

Revised:

After baseline assessment, eligible patients with moderate to severe OSA will be randomly and equally allocated (the randomization is 1:1 ratio) to the CPAP plus BSC group or BSC only group

Original:

The central random allocation sequence will be produced using the statistical software SAS v.9.4 and managed by an independent statistician blinded to the recruitment, evaluation, and intervention of the participants

Revised:

The central random allocation sequence will be produced using the statistical software SAS v.9.4 and managed by an independent statistician (from Department of Epidemiology, School of Public Health, Shanghai Jiao Tong University) blinded to the recruitment, evaluation, and intervention of the participants

Original:

however, it can be blinded to outcome assessors and data statisticians.

Revised:

however, it can be blinded to outcome assessors and data statisticians, they will be blinded to the patients' group assignments (i.e., nobody will tell them which patient is allocated to CPAP treatment group or BSC group).

Comment 7. Page 11 line 46: "standard clinical practice" needs to be defined. What is standard practice? Will the study investigators be checking the fidelity and compliance to this standard practice?

Response 7: Thank you very much for your comment. The standard clinical practice is: before CPAP treatment, doctors will teach the patients how to use the CPAP machine and comfort them. Before allocation, each patient will use CPAP one night, if the patient who is not tolerant to CPAP treatment, we will not include him. Through a chip in this machine, we can monitor usage of CPAP from every participant in our computer terminal through wireless network and might be helpful for improving compliance of CPAP by guidance and assistance.

Original:

CPAP treatment (LOTUS AUTO; Curative Medical Technology Inc., Beijing, China) will be initiated using standard clinical practice at each center

Revised:

CPAP treatment (LOTUS AUTO; Curative Medical Technology Inc., Beijing, China) will be initiated using standard clinical practice (Before CPAP treatment, the standard clinical practice is the doctors will teach the patients how to use the CPAP machine and comfort them) at each center. Through a chip in this machine, we can monitor usage of CPAP from every participant in our computer terminal through wireless network and might be helpful for improving compliance of CPAP by guidance and assistance.

Comment 8. Page 12 line 5: What is auto CPAP?

Response 8: Thank you very much for your comment. Auto CPAP (auto-adjusting CPAP) changes the pressure delivered depending on the residual events detected at any given time, and applies the lowest effective pressure. The average overnight applied pressure is significantly lower with Auto CPAP, while comparably low apnea + hypopnoea indices are achieved. Theoretically, when using Auto CPAP for long-term treatment, the varying pressure delivered would improve patient comfort and thus enhance compliance. (Pepin, J.L., et al., Thorax, 2016. 71: p. 726-733).

Comment 9. Page 12 line 51: additional ")" after brain regions

Response 9: Thank you very much for your comment. We deleted the additional ")" as you suggested.

Comment 10. Page 13 line 27: typo for "approch"

Response 10: Thank you very much for your comment. We revised the "approch" to "approach" as you suggested.

Comment 11. Page 18 safety measurements: Is there a data monitoring committee for this trial?

Response 11: Thank you very much for your comment. Yes, we have a data monitoring committee. The Ethics Committee of Shanghai Jiao Tong University Affiliated Sixth People's Hospital will monitor the data we submit.

Original:

Safety measurements

None

Revised:

Safety measurements

Data we submit will be monitored by Ethics Committee of Shanghai Jiao Tong University Affiliated Sixth People's Hospital.

Comment 12. Page 18 sample size calculation: How was the sample size of 74 for the health controls derived?

Response 12: Thank you very much for your comment. The ratio of cases (patients with OSA, n=148) and health controls was 2:1, but according to Professor Craig L Phillips's suggestion, we deleted the case-control design.

Comment 13. Page 21: Could the authors discuss the pros and cons of not using sham CPAP in this study?

Response 13: Thank you very much for your comment. Sham devices have been validated as a placebo for CPAP, but there is no consensus on the ideal comparator. In this study, we will not use the sham CPAP for the following reasons: both sham CPAP and the absence of treatment for OSA are considered to be adequate controls for an active CPAP intervention. In addition, sham CPAP is not a perfect placebo, since it may result in significant reductions in the number of apnea events, increases in the number of hypopnea events, and a small impairment in sleep quality (Djavadkhani Y, et al. Thorax 2015;0:1–5.).

Comment 14. The SPIRIT checklist needs to be attached.

Response 14: Thank you very much for your comment. We attached the SPIRIT checklist as you suggested. We ensured that all points of SPIRIT check-list are included in this manuscript.

Reviewer: 2

Reviewer Name: Craig L Phillips

Institution and Country: University of Sydney, Australia

Please state any competing interests: None declared

Please leave your comments for the authors below

This proposed multi-center RCT aims to determine whether and when neurocognitive architecture and function in OSA patients can be improved by CPAP treatment.

The authors hypothesise that 3 months of CPAP treatment of OSA will result in improved cognitive function through “structural and functional brain modulation” and that this improvement will continue at 6 and 12 month follow-ups. Hence the primary aim is to determine whether CPAP treatment improves neurocognitive function and “neurocognitive architecture”. A number of secondary aims include investigating whether changes in sleep parameters are associated with cognitive function, neurocognitive architecture and function and whether metabolomics and gut microbiota changes have beneficial effects on neurocognitive impairment.

The 5 center study plans to adopt a prospective randomized open label blinded endpoint (PROBE) design and intends to follow both SPIRIT and CONSORT statements. The interventions are CPAP + Best Supportive Care (BSC) or BSC alone. All outcomes will be assessed at Baseline, 3, 6 and 12 months. A control group who are presumably OSA free will also be enrolled.

Questions and comments for the investigators:

Comment 1. Study design: What is the rationale for embedding a case-control study in this RCT?

Response 1: Thank you very much for your comment. Comparing patients with OSA to healthy controls will allow us to identify differences in neurocognitive function between the two groups. Thus, the rationale for embedding a case-control study in this RCT is to see the differences of primary and secondary outcomes between OSA and controls. However, when we carefully considering your comments, we agreed that the case-control design is redundant, thus we delete the case-control design part.

Comment 2. Study design: Only patients who are CPAP compliant (inclusion criteria) will be included. It was unclear when compliance was to be determined – before or after randomization? If determined before, then will patients be washed out prior to randomizing? If after, then this will introduce bias into the study since CPAP compliance may also be a surrogate for compliance with other medications and lifestyle practices.

Response 2: Thank you very much for your comment. We are sorry we did not state clearly the compliance will be determined before randomization. Before randomization, we will let patients use CPAP one night, if the patient does not tolerate CPAP treatment; we will exclude the above-mentioned patients. The patients who receive one-night CPAP treatment will be washed out (one night) prior to randomizing.

Original:

Inclusion criteria:

Adherent to CPAP treatment

Revised:

Inclusion criteria:

Be tolerate to CPAP treatment

Comment 3. Diagnostic criteria: AASM defined AHI is mentioned – please reference which AASM criteria will be used?

Response 3: Thank you very much for your comment. We used AASM 2007 criteria (Iber C, Chesson A, Quan SF for the American Academy of Sleep Medicine. The AASM manual for the scoring of sleep and associated events: rules, terminology and technical specifications. Westchester, IL: American Academy of Sleep Medicine, 2007) to define AHI, and we added the reference in this manuscript as you suggested.

Original:

The definition of OSA is established according to the AASM criteria.

Revised:

The definition of OSA is established according to the AASM 2007 criteria.

Comment 4. Inclusion criteria: Age 30-65 – please explain why adults younger than 30 and older than 65 are excluded?

Response 4: Thank you very much for your comment. Before we design this RCT, we found patients with OSA in the Shanghai Sleep Health Study (SSHS) of our sleep center (Guan J, Yin S et al. Thorax. 2016;71(4):347-55.) aged mostly between 30 and 65 years old. That's why we choose this age group. Narrowing the age range may reduce the heterogeneity in our study.

Comment 5. Inclusion criteria: Newly diagnosed OSA – do you mean newly diagnosed severe OSA? By excluding moderate OSA the findings will be less generalizable.

Response 5: Thank you very much for your comment. Different from in west countries, in China, fewer OSA patients will use CPAP as CPAP is expensive when compared with average salaries, and the government health insurance does not cover the fee of CPAP. That's why we first want to only include severe OSA because these patients are more eager for treatment. However, we think less generalizable findings caused by excluding moderate OSA is more notable and should be avoided for a RCT, thus we will include newly diagnosed moderate to severe OSA as you suggested.

Comment 6. Inclusion criteria: Adherence to CPAP treatment – what is considered adherent?

Response 6: Thank you very much for your comment.

Before randomization, we will let patients try to use CPAP one night, if the patient does tolerate CPAP treatment; we will consider this patient as adherent. We are sorry we did not state this clearly.

Original:

Inclusion criteria:

Adherent to CPAP treatment

Revised:

Inclusion criteria:

Be tolerate to CPAP treatment

Comment 7. Exclusion criteria: Sleep disorders - Central sleep apnea – please define what proportion of the AHI needs to be central before patients are excluded?

Response 7: Thank you very much for your comment. If the proportion of AHI of central sleep apnea is no less than 50% of the whole AHI, then it defines central sleep apnea (American Academy of Sleep Medicine. International Classification of Sleep Disorders, 3rd ed[M]. Darien: American Academy of Sleep Medicine, 2014)

Comment 8. Exclusion criteria: Alcoholism – please define?

Response 8: Thank you very much for your comment. Alcoholism refers to both alcohol abuse and alcohol dependence. According to National Institute on Alcohol Abuse and Alcoholism, alcoholism in men was defined as alcohol consumption exceeding 14 standard drinks per week, or 4 drinks per day; in women was defined as > 7 standard drinks per week or 3 drinks per day. A standard drink was defined as one 12-ounce bottle of beer, one 5-ounce glass of wine, or 1.5 ounces of distilled spirits (<http://pubs.niaaa.nih.gov/publications/aa68/aa68.htm>; <http://pubs.niaaa.nih.gov/publications/arh27-1/5-17.htm>).

Comment 9. Exclusion criteria: Sedatives – is this present use or any (past or present) use?

Response 9: Thank you very much for your comment. We excluded the sedatives in the present use for it is affecting PSG monitoring.

Comment 10. Exclusion criteria: Deemed by the researchers to be “suitable” for this trial – do you

mean unsuitable? Will you include commercial drivers or people at risk of driving related accidents?
Response 10: Thank you very much for your comment. We made a mistake in this manuscript, and we revised "suitable" to "unsuitable" correctly. The commercial drivers or people at risk of driving related accidents will be unsuitable, because including them will affect the results of neurocognitive function in our study.

Comment 11. Withdrawal criteria: Adverse events related to CPAP – please define?

Response 11: Thank you very much for your comment. We think “Adverse events related to CPAP” is an inappropriate statement. It should be changed to “insurmountable claustrophobia or other unbearable conditions of CPAP intolerance (includes blocked up nose, runny nose, nose bleed, dry throat, irritated eyes, irritated bowl, transient deafness, increased awakenings, uncomfortable pressure of the mask, mask leaks cold air, disturbing noise, problems of exhaling and anxiety during treatment)” (Brostrom, A., et al., J Sleep Res, 2010. 19: p. 603-611.)

Original:

Withdrawal criteria and management:

Subjects will be allowed or required to withdraw from this trial for Adverse events related to CPAP,

Revised:

Withdrawal criteria and management:

Subjects will be allowed or required to withdraw from this trial for insurmountable claustrophobia or other unbearable conditions of CPAP intolerance,

Comment 12. Intervention: CPAP treatment will be initiated using “standard clinical practice”. Please clarify what this involves?

Response 12: Thank you very much for your comment. Before CPAP treatment, the standard clinical practice is the doctors will teach the patients how to use the CPAP machine and comfort them. Before allocation, each patient will use CPAP one night, if the patient who is not tolerant to CPAP treatment, we will not include him. Through a chip in this machine, we can monitor usage of CPAP from every participant in our computer terminal through wireless network and might be helpful for improving compliance of CPAP by guidance and assistance.

Original:

CPAP treatment (LOTUS AUTO; Curative Medical Technology Inc., Beijing, China) will be initiated using standard clinical practice at each center

Revised:

CPAP treatment (LOTUS AUTO; Curative Medical Technology Inc., Beijing, China) will be initiated using standard clinical practice (Before CPAP treatment, the standard clinical practice is the doctors will teach the patients how to use the CPAP machine and comfort them) at each center. Through a chip in this machine, we can monitor usage of CPAP from every participant in our computer terminal through wireless network and might be helpful for improving compliance of CPAP by guidance and assistance.

Comment 13. Intervention: How will the effectiveness of CPAP treatment be determined?

Response 13: Thank you very much for your comment. We determined the effectiveness of CPAP treatment as the use of the CPAP machine for no less than 4 hours per night for at least 70% of the days monitored (Kribbs NB, et al. Am Rev Respir Dis 1993;147: 887-895).

Comment 14. Sleep variables: Why are there no sleep parameters being measured – eg total sleep time, total REM sleep time and total slow wave sleep time.

Response 14: Thank you very much for your comment. We only listed the typical sleep parameters. Standard PSG will be used in this RCT, thus sleep parameters such as total sleep time, total REM sleep time and total slow wave sleep time will also be measured. We added these sleep parameters in this manuscript as you suggested.

Original:

Sleep variables (apnea-hypopnea index [AHI], oxygen desaturation index [ODI], lowest oxygen saturation [LSaO₂]) mean oxygen saturation [mean SaO₂], micro-arousal index [MAI])

Revised:

Sleep variables (apnea-hypopnea index [AHI], oxygen desaturation index [ODI], lowest oxygen saturation [LSaO₂]) mean oxygen saturation [mean SaO₂], micro-arousal index [MAI], total sleep time, total REM sleep time and total slow wave sleep time, etc) .

Comment 15. Outcome Assessment: (page 12, line 21) – the staff will be “blinded to the allocation results of participants”. Do you mean staff will be blinded to participant treatment allocation?

Response 15: Thank you very much for your comment. Yes, the staffs assessing outcomes will be blinded to participant treatment allocation.

Comment 16. Primary outcome: Please clarify what the primary outcome is? According to the manuscript, several are listed (MoCA and MMSE as well as “structure” and “function” of related brain regions using fMRI - where neither structure nor function is clearly defined).

Response 16: Thank you very much for your comment. The primary outcome is MoCA. We moved “MMSE questionnaire” and “structure and function of related brain regions will be measured using fMRI” to the secondary outcome. The neuropsychological evaluation and a functional and structural magnetic resonance imaging study of connectivity within the default mode network (DMN) and cortical thickness (CTh). We revised the primary and secondary outcomes in this manuscript accordingly.

Comment 17. Primary outcome: It is also unclear whether this list of outcomes is a composite outcome or not. If it is a composite outcome then please define how it will be determined?

Response 17: Thank you very much for your comment. The list of outcomes is not a composite outcome. The list of outcomes is all used to assess the neurocognitive architecture and function, include: 1) MoCA and MMSE: assess cognitive function; 2) The neuropsychological evaluation and a functional and structural magnetic resonance imaging study of connectivity within the default mode network (DMN) and cortical thickness (CTh). We revised the primary and secondary outcomes in this manuscript according to your advice.

Comment 18. Secondary outcome: Please describe with references the Metagenomics and Metabolomics analysis and what will be quantified?

Response 18: Thank you very much for your comment. Metabolomics analysis will quantify the characteristic changes in low-molecular-weight (< 1000) metabolites in a pathophysiological state (Xu H, Yin S et al. Sci Rep. 2016;6:30958); Metagenomics analysis will quantify microbial community profiling in the gut (Zhang C, et al. EBioMedicine. 2015 Jul 10;2(8):968-84.).

Comment 19. Brain imaging: The past tense is used as if this imaging has already taken place? For example: “MRI scan was conducted”, “24 functional images were obtained”. Please clarify?

Response 19: Thank you very much for your comment. The imaging has not taken place, we revised it as “MRI scan will be conducted”, “24 functional images will be obtained”.

Comment 20. Cerebral metabolites: The same past tense “scans were performed”.

Response 20: Thank you very much for your comment. We have revised it as “scans will be performed”.

Comment 21. Abdominal and liver fat: “were measured”.

Response 21: Thank you very much for your comment. We have revised it as “Abdominal and liver fat will be measured”

Comment 22. Abdominal fat: There is no mention of where the images will be scanned in relation to bony landmarks – for example abdominal fat might be quantified from contiguous slices starting at the

L4/5 intervertebral space. Please clarify?

Response 22: Thank you very much for your comment. We consult the radiologists in our hospital again, they measure the hepatic, subcutaneous fat and visceral fat will through axial scanning in upper abdomen (from diaphragm to the lower edge of liver). They measure the abdominal fat from diaphragm to the lower edge of liver, not according to the bony landmarks.

Comment 23. Statistics: It is unclear what statistical analysis will be performed to assess primary and secondary outcomes.

Response 23: Thank you very much for your comment. To assess primary and secondary outcomes, the mixed-effects linear model with restricted maximum likelihood will be used to analyze the effect of CPAP treatment on the continuous variables, and logistic regression models for dependent categorical variables. In each model, the independent variables are group (CPAP plus BSC group, BSC group), different measurement time (e.g., baseline, 3, 6 and 12-month follow-up), confounding factors (e.g., age, BMI) and any inequality factors among groups at the baseline assessment. The post hoc comparison between groups at different time points will be performed using multiple comparisons.

Original:

Statistical methods

None

Revised:

Statistical methods

To assess primary and secondary outcomes, the mixed-effects linear model with restricted maximum likelihood will be used to analyze the effect of CPAP treatment on the continuous variables, and logistic regression models for dependent categorical variables. In each model, the independent variables are group (CPAP plus BSC group, BSC group), different measurement time (e.g., baseline, 3, 6 and 12-month follow-up), confounding factors (e.g., age, BMI) and any inequality factors among groups at the baseline assessment. The post hoc comparison between groups at different time points will be performed using multiple comparisons.

Comment 24. Discussion: It is mentioned that the investigators will determine whether gut microbiota “play a pivotal role in OSA”. This implies that alterations in microbiota somehow cause OSA? Please explain.

Response 24: Thank you very much for your comment. Based on previous rodent experiments, we think that alterations in microbiota might cause OSA-related cardiovascular and metabolic complications to some extent, even neurocognitive dysfunction, not OSA itself. We reorganized this statement more clearly.

Original:

we will determine whether gut microbiota play a pivotal role in OSA and neurocognitive dysfunction

Revised:

we will determine whether gut microbiota play a pivotal role in OSA-related complications, especially in alleviating neurocognitive dysfunction

Comment 25. Discussion: “The results of this study will help to clarify potential mechanisms for alleviation of OSA” – please explain how this might occur?

Response 25: Thank you very much for your comment. We speculate that if proportion of beneficial bacteria increased, the OSA-related cardiovascular and metabolic status will improve. Thus, the results of this study will help to clarify potential mechanisms for alleviation of OSA-related complications.

Original:

Discussion:

None

Revised:

Discussion:

The microbial community structure in rodent animals was changed when the rodents were under intermittent hypoxia or sleep fragmentation status. In addition, sleep fragmentation-induced metabolic alterations may be mediated by concurrent changes in gut microbiota. Thus, gut microbiome-targeted therapeutics may help to reduce the major end-organ morbidities of OSA.

Summary of main points:

Comment 26: Overall this is an interesting study but relatively ambitious in regards to the large number of investigations and outcomes that are to be collected at multiple time-points. There will likely be a large burden on patients and one wonders whether the projected 20% loss to follow-up is realistic.

Response 26: Thank you very much for your comment. The multi-center clinical research project from school of medicine, Shanghai Jiao Tong University (DLY201502) supported a large sum of money, and five sleep centers will be participated in this study. Though it is a big project, we have enough funds and staffs to enable us to complete this multi-center RCT. Meanwhile, we will inform the participants that we will reward them with CPAP machine if they finish the projects, this will enhance their compliance. We will try our best to conduct this multi-center RCT, and try to make 20% or less loss to follow-up realistic.

Comment 27: It is unclear why the investigators have opted to embed a case-control study into this randomized controlled trial.

Response 27: Thank you very much for your comment. We have answered this question in Comment 1.

Comment 28: It is of concern that only patients who are compliant with CPAP will be eligible as this will introduce bias if compliance is assessed post randomization.

Response 28: Thank you very much for your comment. We did not state this clearly in previous manuscript. For detail, please see comment 2.

Comment 29: It is unclear how the primary outcome with multiple variables is to be assessed.

Response 29: Thank you very much for your comment. We have answered this question in Comment 16 and 17.

Comment 30: The age range and OSA severity of patients will limit generalizability of findings.

Response 30: Thank you very much for your comment. We have answered this question in Comment 4.

Reviewer: 3

Reviewer Name: Caroline Mitchell

Institution and Country: Academic Unit of Primary Medical Care, Faculty of Medicine, Dentistry and Health,

University of Sheffield, United Kingdom

Please state any competing interests: None declared

Please leave your comments for the authors below

Comment 1: This protocol paper fulfils the criteria for reporting randomised controlled trials: a CONSORT flow chart will be generated. Analysis must state 'intention to treat analysis', which it does not currently.

Response 1: Thank you very much for your comment. 1) We uploaded a CONSORT flow chart as you suggested. 2) We emphasized in the statistical methods part that Study analysis will be performed by

intention-to-treat (ITT). Also, we described the statistical methods in a more detailed way.

Comment 2: I would comment that this study is collecting an extraordinary range of secondary outcome data and I would be concerned about participant outcome measurement burden. In order to address this, it would be helpful to know which of the outcome measures are actually part of routine care in this healthcare context and which are being performed as additional research measures. For example, I am concerned about the additional 'study within a study': the exploration of whether 'CPAP modulates the gut microbiota and contributes to alleviation of neurocognitive impairment. We hypothesize that patients with OSA who undergo CPAP treatment will have metabolic improvement and modulation of gut microbiota, and that these changes will have beneficial effects on neurocognitive impairment'.

Response 2: Thank you very much for your comment. This study is supported by multi-center clinical research project from school of medicine, Shanghai Jiao Tong University (DLY201502), and five sleep centers will be participated in this study. Though it is a big project, we have enough funds and staff to complete this multi-center RCT. Though the outcomes of this study seem too many, we could roughly divided into two parts: healthcare context (PSG monitoring, MRI scan, Blood pressure, Anthropometric variables, Echocardiography, Arterial stiffness, Lung function, Electrocardiography) and research measures (Questionnaires, several serum biomarkers, metabolomics and metagenomics profiling). In our opinion, we think these variables are related to cognitive dysfunction in OSA to some extent, though some connections might be weak. Subjects in our study will only provide urine, venous blood, faeces and participate in several examinations, and these will be bearable.

Comment 3: As a secondary outcome arm of the proposed main study, this would be theory generating and the study is not powered to assess any such associations. I would therefore consider this part of the protocol to be over and above the outcome measures needed to answer the main research question. Whilst the study protocol authors may have funding in place for this additional work, I do not consider that this arm of the study should necessarily be published as integrated within the main study protocol.

Response 3: Thank you very much for your comment. We thought that the secondary outcome arm of this protocol is related to the main outcomes, though we might not get a causal relationship between these two outcome measures (i.e., main outcomes and secondary outcomes). We thought that changes in human intestinal microbiota might help to explain part of the neurological dysfunction in OSA and this idea is very novel. For us, conducting a multicenter RCT is very laborious, time-consuming and costly, thus we beg you to understand our approach in collecting so many data.

We have rechecked the references, figures, and text carefully. We hope that our reply to the reviewer's comments is sufficient. Thank you very much for reconsidering our revised manuscript for publication.

VERSION 2 – REVIEW

REVIEWER	Lian Leng Low Singapore General Hospital, Singapore Health Services, Singapore
REVIEW RETURNED	20-Jan-2017

GENERAL COMMENTS	The authors have adequately addressed my previous comments.
---

REVIEWER	Craig L Phillips University of Sydney, Australia
REVIEW RETURNED	22-Jan-2017

GENERAL COMMENTS	The manuscript has been greatly improved including by deleting the case-control design from the protocol. I think it would still benefit overall by having someone with a better command of the English language helping to correct word choices and grammar. For example:  1. Strengths/Limitations (Editorial Requirements) Possible selection bias “may be existed” should read “may exist”. 2. insurmountable claustrophobia or other unbearable conditions of CPAP intolerance should be “or other unbearable conditions associated with CPAP use” 3. Also, doctors will teach patients how to use the CPAP machine and “comfort them”. Perhaps, “Doctors will educate patients on the use of CPAP and provide positive encouragement”. 4. Through a chip in this machine, we can monitor usage of CPAP from every participant in our computer terminal through wireless network and might be helpful for improving compliance of CPAP by guidance and assistance. This seems to indicate “online monitoring of CPAP compliance. This information can also be used to improve compliance” 5. Inclusion criteria “Be tolerate to CPAP treatment – should be “Be tolerant of CPAP treatment based on 1 night of use” – one then wonders what tolerant means here – is it based on a patient saying they can tolerate CPAP, and/or do they need to use it for a certain number of hours? These corrections could easily be managed by the post-production editors. Other comments in response to my original review: Comment 7 – Exclusion criteria for central sleep apnea: “>=50% of AHI composed of central apneas” should be added to the definition. Comment 8 – please reference in the manuscript the alcoholism definition that you provided. Comment 9 – please state “current sedative use” Comment 10 – please include “commercial drivers or people deemed to be at risk of driving related accidents” Comment 11 – withdrawal criteria please change the word CPAP “intolerance” to CPAP “use” Comment 13 – CPAP “effectiveness” is a measure of OSA alleviation not a measure of good compliance as defined by >4hrs for 70% of night. I wanted to know how you can determine whether the machine is adequately correcting the OSA as some auto CPAP machines fail to control apneas in some patients.
--

REVIEWER	Caroline Mitchell Academic Unit of Primary Medical Care University of Sheffield, Sheffield S5 5AU UK
REVIEW RETURNED	16-Jan-2017

GENERAL COMMENTS	The authors have satisfactorily addressed the feedback which I gave on the earlier version of the manuscript in this revised version (major revisions have been undertaken). The main document now clearly states primary and secondary endpoints; the abstract would be improved by including these primary and secondary endpoints.
---

	A CONSORT flow chart has now been attached. In the statistical methods it is now stated that analysis will be performed by intention-to-treat (ITT) analysis. The study still includes an extraordinary range of secondary outcome data but the resources and actions to minimise participant burden (part of routine care assessment) is emphasised in the authors responses and within the revised version of the manuscript. The authors make it more clear in this revised manuscript paper, that the secondary outcome data collection relating to changes in human intestinal microbiota would be theory generating and that the study is not powered to establish any causal associations.
--	---

VERSION 2 – AUTHOR RESPONSE

Reviewer: 1

Reviewer Name: Lian Leng Low

Institution and Country: Singapore General Hospital, Singapore Health Services, Singapore

Please state any competing interests: None declared

Please leave your comments for the authors below

The authors have adequately addressed my previous comments.

Response: Thank you very much for your high enthusiasm and comprehensive analysis of our manuscript.

Reviewer: 2

Reviewer Name: Craig L Phillips

Institution and Country: University of Sydney, Australia

Please state any competing interests: None declared

Please leave your comments for the authors below

The manuscript has been greatly improved including by deleting the case-control design from the protocol. I think it would still benefit overall by having someone with a better command of the English language helping to correct word choices and grammar. For example:

Comment 1. Strengths/Limitations (Editorial Requirements) Possible selection bias “may be existed” should read “may exist”.

Comment 2. insurmountable claustrophobia or other unbearable conditions of CPAP intolerance should be “or other unbearable conditions associated with CPAP use”

Comment 3. Also, doctors will teach patients how to use the CPAP machine and “comfort them”.

Perhaps, “Doctors will educate patients on the use of CPAP and provide positive encouragement”.

Comment 4. Through a chip in this machine, we can monitor usage of CPAP from every participant in our computer terminal through wireless network and might be helpful for improving compliance of CPAP by guidance and assistance. This seems to indicate “online monitoring of CPAP compliance. This information can also be used to improve compliance”

Comment 5. Inclusion criteria “Be tolerate to CPAP treatment – should be “Be tolerant of CPAP treatment based on 1 night of use” – one then wonders what tolerant means here – is it based on a patient saying they can tolerate CPAP, and/or do they need to use it for a certain number of hours? These corrections could easily be managed by the post-production editors.

Response 1~5: Thank you very much for your high enthusiasm and comprehensive analysis of our manuscript. We revised the comments you mentioned-above. In addition, this manuscript has been checked for language and grammar by at least two professional copy-editors who are native speakers of English (by a professional English editing company-textcheck). For a certificate, please see: <http://www.textcheck.com/certificate/ySqB6H>.

Other comments in response to my original review:

Comment 6 – Exclusion criteria for central sleep apnea: “ $\geq 50\%$ of AHI composed of central apneas” should be added to the definition.

Response 6: Thank you very much for your comment. We have added a definition of central sleep apnea to this manuscript, as suggested.

Comment 7 – please reference in the manuscript the alcoholism definition that you provided.

Response 7: Thank you very much for your comment. We have added a reference regarding the definition of alcoholism, as suggested.

Comment 8 – please state “current sedative use”

Response 8: Thank you very much for your comment. We have used “current sedative use” in the revised version.

Comment 9 – please include “commercial drivers or people deemed to be at risk of driving related accidents”

Response 9: Thank you very much for your comment. We have added “commercial drivers or people deemed to be at risk of driving-related accidents” to the exclusion criteria.

Comment 10 – withdrawal criteria please change the word CPAP “intolerance” to CPAP “use”

Response 10: Thank you very much for your comment. As suggested, we have changed the word CPAP “intolerance” to CPAP “use”.

Comment 11 – CPAP “effectiveness” is a measure of OSA alleviation not a measure of good compliance as defined by >4 hrs for 70% of night. I wanted to know how you can determine whether the machine is adequately correcting the OSA as some auto CPAP machines fail to control apneas in some patients.

Response 11: Thank you very much for your comment. The auto CPAP device we purchased can record apnea events, and using a chip in this machine, we will monitor the use of CPAP (whether effective or not) in every participant via a wireless network.

Reviewer: 3

Reviewer Name: Caroline Mitchell

Institution and Country: Academic Unit of Primary Medical Care, University of Sheffield, Sheffield, S5 5AU, UK

Please state any competing interests: None declared

Please leave your comments for the authors below

The authors have satisfactorily addressed the feedback which I gave on the earlier version of the manuscript in this revised version (major revisions have been undertaken).

The main document now clearly states primary and secondary endpoints; the abstract would be improved by including these primary and secondary endpoints.

A CONSORT flow chart has now been attached.

In the statistical methods it is now stated that analysis will be performed by intention-to-treat (ITT) analysis.

The study still includes an extraordinary range of secondary outcome data but the resources and actions to minimise participant burden (part of routine care assessment) is emphasised in the authors responses and within the revised version of the manuscript.

The authors make it more clear in this revised manuscript paper, that the secondary outcome data

collection relating to changes in human intestinal microbiota would be theory generating and that the study is not powered to establish any causal associations.

Response: Thank you very much for your high enthusiasm and comprehensive analysis of our manuscript.

We have rechecked the references, figures, and text carefully. We hope that our reply to the reviewer's comments is sufficient. Thank you very much for reconsidering our revised manuscript for publication.

VERSION 3 – REVIEW

REVIEWER	Craig L Phillips University of Sydney, Australia
REVIEW RETURNED	27-Feb-2017

GENERAL COMMENTS	I believe the authors have addressed all my concerns.
---